# EVIAN: TOWARDS EXPLAINABLE
# VISUAL INSTRUCTION-TUNING DATA AUDITING

## ABSTRACT

The efficacy of Large Vision-Language Models (LVLMs) is critically dependent on the quality of their training data, requiring a precise balance between visual fidelity and instruction-following capability. Existing datasets, however, are plagued by inconsistent quality, and current data filtering methods rely on coarse-grained scores that lack the granularity to identify nuanced semantic flaws like logical fallacies or factual errors. This creates a fundamental bottleneck in developing more reliable models. To address this, we make three core contributions. First, we construct a large-scale, 300K-sample benchmark by systematically injecting diverse, subtle defects to provide a challenging testbed for data auditing. Second, we introduce a novel "Decomposition-then-Evaluation" paradigm that breaks model responses into constituent cognitive components: visual description, subjective inference, and factual claim, enabling targeted analysis. Third, we instantiate this paradigm via **EVIAN** (**E**xplainable **V**isual **I**nstruction-tuning Data **A**uditi**N**g), a pipeline that evaluates these components along the orthogonal axes of Image-Text Consistency, Logical Coherence, and Factual Accuracy. Our empirical findings challenge the prevailing scale-centric paradigm: a model fine-tuned on a compact, high-quality subset curated by EVIAN consistently surpassed models trained on orders-of-magnitude larger datasets. We also reveal that dividing complex auditing into verifiable subtasks enables robust curation, and that Logical Coherence is the most critical factor in data quality evaluation.

## 1 INTRODUCTION

Large Vision-Language Models (LVLMs) (Chen et al., 2024e) have recently demonstrated remarkable progress in aligning visual perception with natural language understanding, enabling a wide range of applications from medical assistance to robotic control (Yin et al., 2024). An important factor of this success is *Visual Instruction Tuning* (VIT), which aligns visual representations with language instructions to enhance instruction-following capability (Liu et al., 2023). However, the effectiveness of VIT hinges on the quality of the underlying training data, which must strike a delicate balance between adhering to user commands and maintaining fidelity to visual inputs.

Existing datasets and filtering methods fall short of this requirement. Large-scale data synthesis (e.g., LLaVA-Instruct-150K ) improves instruction following but often introduces noise (Liu et al., 2024c; Tang et al., 2024), while similarity-based filtering methods (e.g., CLIP score) promote visual grounding but lack the granularity to detect subtle semantic flaws (Wang et al., 2024a). As a result, current LVLMs frequently suffer from fine-grained errors, including object hallucination, attribute misattribution, factual inconsistency, and flawed reasoning (Liu et al., 2024a; Bai et al., 2024; Chen et al., 2024d). These deficiencies reveal a fundamental bottleneck: prevailing approaches rely on coarse, uni-dimensional quality measures that collapse diverse error types into a single opaque score.

In this work, we argue that evaluating model-generated responses requires moving beyond monolithic scoring toward structured verification. Our core insight is that a response is not an indivisible block of text but a composite of distinct, verifiable components. Building on this principle, we propose the *Decomposition-then-Evaluation* paradigm, which reframes the task of auditing complex responses into targeted sub-tasks. Specifically, we isolate and validate *pure visual descriptions* to address visual misrepresentation, *external factual claims* to correct factual inaccuracies, and *subjective inferences* to mitigate flawed reasoning.

Figure 1: Illustration of the core challenge in Visual Instruction Tuning (VIT), showing positive examples (middle) and negative examples (right).

To operationalize this paradigm, we introduce **EVIAN** (**E**xplainable **V**isual **I**nstruction-tuning Data **A**diti**N**g), an automated and interpretable framework that systematically evaluates responses along three orthogonal axes: Image-Text Consistency, Logical Coherence, and Factual Accuracy. Complementing this framework, we construct a large-scale, 300K-sample benchmark by injecting diverse, subtle defects, providing a challenging testbed for fine-grained data auditing. Our empirical findings show that models fine-tuned on compact, high-quality subsets curated by EVIAN consistently outperform models trained on orders-of-magnitude larger datasets, highlighting that interpretable data curation, rather than sheer scale, is the key to advancing LVLMs.

Our main contributions are as follows:

- To spur research in LVLM visual instruction tuning data quality and facilitate rigorous evaluation, we introduce a 300K-sample benchmark for visual instruction data selection, built by systematically injecting diverse semantic defects to support fine-grained auditing.

- We propose the *Decomposition-then-Evaluation* paradigm and instantiate it in **EVIAN**, a fully automated and interpretable framework that decomposes responses into visual descriptions, subjective inferences, and factual claims, and evaluates them along three orthogonal dimensions.

- We conduct extensive experiments showing that for LVLMs, the logical integrity of training data is a more decisive factor for downstream performance than its informational richness, establishing the critical need to prioritize reasoning and factual correctness in data curation.

## 2 RELATED WORK

The evolution of vision-language data curation from coarse pre-training filters to more nuanced instruction tuning methods reveals a persistent bottleneck: the lack of scalable, fine-grained evaluation. This forces a reliance on shallow quality proxies, hindering the development of more trustworthy models.

**Data Selection for Vision-Language Pre-training.** The initial challenge in vision-language learning is distilling high-quality subsets from noisy web-crawled datasets like LAION (Schuhmann et al., 2021). A dominant strategy uses pre-trained models such as CLIP (Radford et al., 2021), AL-BEF (Li et al., 2021), and BLIP (Li et al., 2022; 2023) for similarity-based filtering (Hessel et al., 2021; Xu et al., 2025a; Wang et al., 2024c), a technique later refined by methods like mixture models (Shi et al., 2024a). Another approach leverages generative models to sanitize datasets by re-captioning or correcting labels (Vasa et al., 2025; Mahjourian & Nguyen, 2025; Zhang et al., 2024; Zhu et al., 2023). However, both paradigms rely on coarse proxies like a single similarity score, lacking the diagnostic insight to avoid erroneously removing valuable, complex samples.

**Data Curation for Visual Instruction Tuning.** As the training objective shifts from broad representation learning to nuanced instruction-following in VIT (Safaei et al., 2025), the need for high-quality data becomes more acute (Chen et al., 2024c). One strategy involves generating synthetic

data (Liu et al., 2024d; Chen et al., 2024a), but this can bypass the complexity of real-world noise. A more prominent approach is curating data via the LLM-as-a-Judge paradigm (Gu et al., 2024; Li et al., 2024; Pu et al., 2025). This method, however, is vulnerable to the cognitive biases, reasoning shortcuts, and instability of LLM judges, especially without ground-truth references (Shi et al., 2024b; Hwang et al., 2025; Ye et al., 2024; Guerdan et al., 2025; Wei et al., 2024). Thus, scalable and trustworthy data curation remains an unsolved problem.

**The Gap in Fine-Grained Evaluation.** The reliance on coarse-grained proxies stems from a deeper challenge: the absence of scalable, fine-grained evaluation. While research has pursued metrics beyond a single holistic score (Adlakha et al., 2024), a framework for deep error diagnosis remains elusive. Early automated methods were constrained by fixed criteria ill-suited for open-ended semantic errors (Zhao et al., 2024). Recent pipelines like SCALE (Xu et al., 2025b) offer a more holistic assessment but still lack a deep focus on compositional logic. Specialized benchmarks for tasks like logical reasoning have been introduced (Xiao et al., 2024; Xu et al., 2025c), but their specialization renders them too narrow for general use. While some work advocates for more comprehensive evaluation (Tu et al., 2025), it often remains conceptual. This void of effective diagnostic tools forces data curation back to shallow logic, perpetuating a cycle that limits model improvement and highlights the urgent need for a new paradigm in fine-grained assessment.

## 3 THE METHOD: EVIAN

We propose **EVIAN**, an automated pipeline for auditing visual instruction data. As illustrated in Figure 2, EVIAN follows a two-phase process: (i) response decomposition, which disentangles complex answers into verifiable components, and (ii) multi-faceted evaluation, which scores these components across orthogonal quality dimensions.

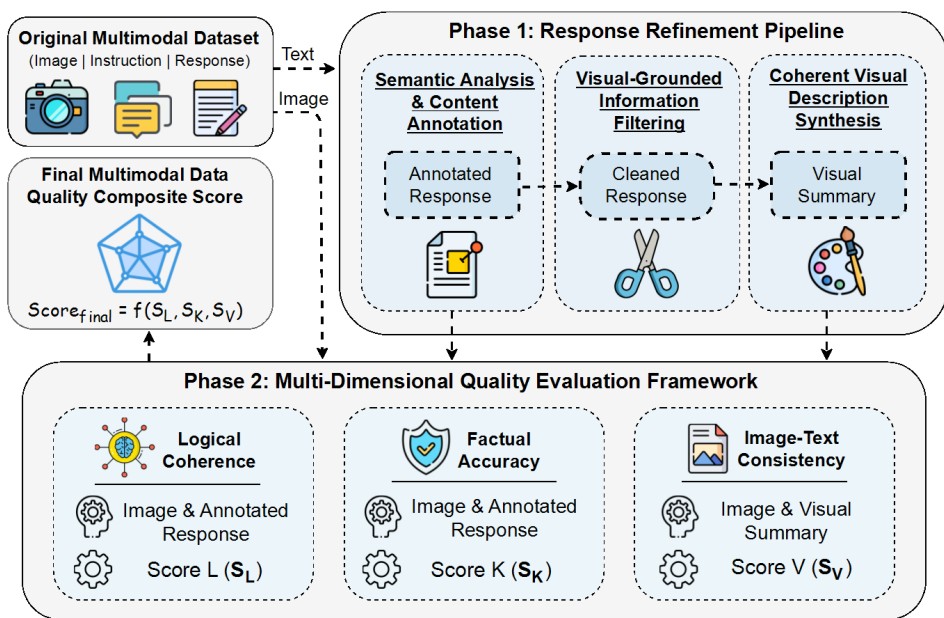

Figure 2: Overview of the EVIAN framework's two-phase process. EVIAN first decomposes a response into its visual, inferential, and factual components, then evaluates them across the orthogonal dimensions of Image-Text Consistency, Logical Coherence, and Factual Accuracy.

### 3.1 PROBLEM DEFINITION AND DATA QUALITY METRICS

We define *visual instruction data auditing* as the task of assigning interpretable quality scores to image-instruction-response triples. Formally, given $x_i = (I_i, P_i, R_i)$ from dataset $D$, our auditing

function $\Phi$ maps each sample to a three-dimensional score vector:

$$\boldsymbol{S}_i = \Phi(x_i) = (S_{L,i}, S_{K,i}, S_{V,i}), \tag{1}$$

where each score ranges from 1 (low) to 5 (high). The three metrics are:

- **Logical Coherence** ($S_L$): soundness of reasoning relative to the instruction and visual evidence.
- **Factual Accuracy** ($S_K$): correctness of knowledge claims against external facts.
- **Image-Text Consistency** ($S_V$): fidelity of the textual response to the visual input.

Together, these axes provide a comprehensive measure of data quality, capturing both semantic integrity and visual fidelity.

## 3.2 PHASE 1: RESPONSE DECOMPOSITION VIA CHAIN-OF-THOUGHT

The first phase disentangles raw responses into verifiable components, separating visual descriptions from subjective inferences and factual claims. This is achieved through a three-step chain-of-thought (CoT) process, $\Psi_{\text{deconstruct}}$, implemented with the Qwen3-235B-A22B-Instruct model (Yang et al., 2025). The result is an annotated response with explicit tags and a purified visual summary, which together form the basis for systematic auditing.

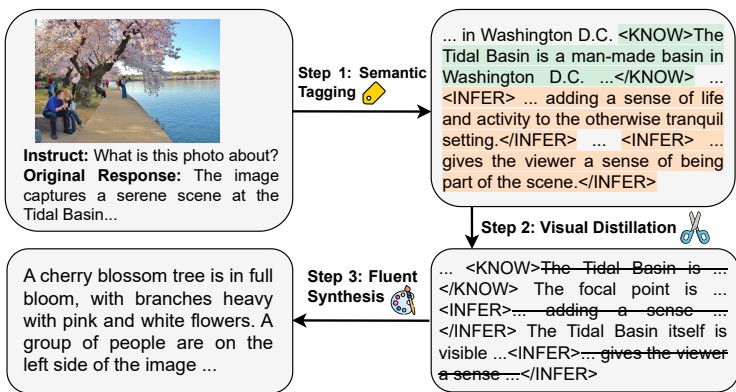

Figure 3: The three-stage Chain-of-Thought (CoT) process for response decomposition. It involves: 1) isolating subjective inferences and factual claims via semantic tagging, 2) purifying the text through visual distillation, and 3) refining the output into a cohesive, purely visual summary.

**Step 1: Semantic Tagging.** The process begins by parsing the raw response $R_i$ while strictly preserving its original wording. Subjective judgments (e.g., "the room feels cozy") are wrapped in `<INFER>` tags, and knowledge-dependent claims (e.g., "this is a Bauhaus-style lamp") are wrapped in `<KNOW>` tags. Untagged text is treated as purely visual description. This produces an annotated response $R_i^{\text{annotated}}$ that explicitly separates cognitive components without altering their content.

**Step 2: Visual Distillation.** Next, the annotated response is distilled into a purely visual form. Segments within `<INFER>` or `<KNOW>` tags are either rewritten into neutral, descriptive statements or deleted if unverifiable. For example, "this is likely a wedding dress" becomes "a white dress"; unverifiable claims are dropped entirely. Untagged visual statements remain unchanged. The result is a draft $R_i^{\text{draft}}$ containing only objective, image-grounded content.

**Step 3: Fluent Synthesis.** Since distillation may fragment the text, a final synthesis step restores fluency and coherence. The draft response is reorganized into a single, natural paragraph while strictly forbidden from adding new content. This ensures the output $R_i^{\text{visual}}$ is a faithful, high-quality visual summary.

Together, these steps yield two complementary artifacts: $R_i^{\text{annotated}}$, which retains the full response structure with explicit tags, and $R_i^{\text{visual}}$, which isolates objective descriptions. This decomposition provides the foundation for precise, component-level auditing in Phase 2.

### 3.3 Phase 2: Multi-faceted Quality Assessment

The second phase conducts a multi-faceted evaluation of each decomposed response along three orthogonal dimensions: logical coherence, factual accuracy, and image-text consistency. We employ Qwen2.5-VL-7B-Instruct (Bai et al., 2025) as an automated auditor, which assigns interpretable 1–5 scores and textual rationales based on a detailed rubric. This step provides fine-grained diagnostics of different error types while producing standardized quality scores that can be aggregated for ranking and selection.

**Logical Coherence** ($S_L$). This dimension evaluates whether reasoning in the `<INFER>` tags follows plausibly from visual evidence. Scores increase with reasoning strength: a default of 2 when no inference is given, 3 for plausible but unsubstantiated claims, 4 for well-supported reasoning, and 5 for logically undeniable conclusions. This rubric rewards depth of reasoning while penalizing speculation.

**Factual Accuracy** ($S_K$). This dimension fact-checks knowledge claims in the `<KNOW>` tags against the auditor's internal knowledge. Fully correct claims receive 5, minor inaccuracies lower the score to 4, and a single major error (e.g., misidentifying a capital city) caps the score at 2. In the absence of knowledge claims, the default score is 2, distinguishing informative from non-informative responses.

**Image-Text Consistency** ($S_V$). This dimension measures the alignment of the purified visual description $R^{\text{visual}}$ with the image. The principle is consistency over completeness: omissions are acceptable, but contradictions or unverifiable assertions are heavily penalized. Perfectly faithful descriptions receive 5, minor imprecisions result in 4, and any clear contradiction drops the score to 2 or below. This ensures that only visually accurate responses achieve the highest marks.

By producing a triplet $(S_L, S_K, S_V)$ with explicit explanations, Phase 2 delivers an interpretable and multi-dimensional quality assessment. These scores directly guide downstream data ranking and selection.

### 3.4 Data Ranking and Selection

To enable downstream filtering, the three-dimensional score vector $\boldsymbol{S}$ is aggregated into a single scalar:

$$S_{\text{overall}} = \frac{S_L + S_K + S_V}{3}. \tag{2}$$

This default scheme assumes equal importance, but weights can be tuned for specific applications. For example, emphasizing $S_K$ for knowledge-intensive tasks or $S_L/S_V$ for creative captioning. This flexibility ensures that data selected by EVIAN aligns with diverse modeling objectives.

## 4 Benchmarking Data Quality via Controlled Defect Injection

To quantitatively validate a data auditing pipeline's ability to detect fine-grained flaws in logical coherence, factual accuracy, and image-text consistency, a tailored benchmark with systematically injected defects is essential, as existing datasets lack the controlled errors needed for such a targeted evaluation. To ensure consistency with prior work, we adopt the SCALE methodology (Xu et al., 2025b) as the starting point for benchmark construction. From its source pool of 500,000 multimodal samples across eight datasets (Table 1), we derive two complementary components: (i) a 50,000-sample "gold standard" set purified by SCALE, and (ii) a 250,000-sample "challenge" set obtained via random down-sampling followed by our defect injection pipeline. Together, these components yield a reproducible benchmark of 300,000 samples, designed to evaluate whether data auditing methods can distinguish clean data from semantically corrupted examples.

**Defect Injection Pipeline.** The challenge set is generated through a three-stage pipeline that leverages the Qwen3-235B-A22B-Instruct model to embed subtle, context-aware flaws. The process is guided by a principled taxonomy (Table 2) spanning three critical dimensions for auditing: *perceptual consistency*, *factual accuracy*, and *logical coherence*.

Table 1: Overview of the eight foundational source datasets forming our comprehensive evaluation pool. This curated selection covers a broad spectrum of tasks, from general VQA to specialized domains like chart interpretation, providing a diverse and challenging testbed for robust multimodal data auditing. (I: Image, T: Text)

| Dataset | Size | Format | Task | Selected size |
|---|---|---|---|---|
| LLaVA-1.5-Mix (Liu et al., 2024b) | 665K | I+T | General QA | 154K |
| ShareGPT-4V (Chen et al., 2024b) | 1.2M | I+T | Caption | 289K |
| Geometry3K (Lu et al., 2021) | 3K | I+T | Mathematics | 466 |
| ChartQA (Masry et al., 2022) | 32K | I+T | Chart | 6K |
| InfoVQA (Mathew et al., 2022) | 30K | I+T | OCR | 5K |
| A-OKVQA (Schwenk et al., 2022) | 24K | I+T | Knowledge | 3K |
| DocVQA (Mathew et al., 2021) | 50K | I+T | Document | 9K |
| AllSeeing-V2 (Wang et al., 2024b) | 127K | I+T | Grounding | 29K |

Table 2: Principled taxonomy of semantic defects, driving our LLM-driven injection pipeline for benchmark construction. These three categories (Image-Text Consistency, Logical Coherence, Factual Accuracy) align with EVIAN's evaluation dimensions, detailing fine-grained subtypes and LLM-guided strategies for embedding subtle, context-aware flaws, crucial for a nuanced data quality testbed.

| Category | Error Subtype | Description / Generation Strategy |
|---|---|---|
| **Consistency** | attribute | Describes an object's attribute incorrectly. |
| | spatial | Details incorrect spatial relations between objects. |
| | action | Assigns a wrong action or state to a subject. |
| | fake | Introduces a plausible yet non-existent object. |
| | misidentification | Misidentifies an existing object. |
| **Reasoning** | conclusion | Generalizes hastily from a single detail. |
| | causal | Mistakes correlation for causation between events. |
| | prediction | Makes a baseless prediction from scant evidence. |
| | procedural | Adds a flawed or superfluous step to a process. |
| | comparison | Forms a misleading analogy from superficial traits. |
| **Knowledge** | entity | Corrupts facts about a named entity. |
| | context | Places an object in a wrong historical/tech context. |
| | definition | Provides an incorrect definition of a concept. |
| | attribution | Misattributes a quote or work to the wrong source. |

**Stage 1: Content Analysis.** Each source response is analyzed by an LLM to identify whether it contains external knowledge or logical reasoning. This structured analysis, output in JSON, serves as a prior to ensure that subsequent errors are coherent with the intrinsic properties of the text.

**Stage 2: Contextual Error Selection.** An error category is chosen via a probabilistic cascade. To counter their rarity, knowledge-related and reasoning-related errors are prioritized with probabilities of 0.8 and 0.6, respectively, while perceptual consistency serves as the default. Subtypes are selected randomly for consistency errors, whereas an additional LLM call determines the most plausible subtype for knowledge and reasoning cases.

**Stage 3: Guided Rewriting.** The chosen error is injected by prompting the LLM with a targeted transformation instruction. A strict system prompt constrains the model to output only the modified text, ensuring automation and reproducibility.

This injection strategy goes beyond simple noise addition: it produces realistic, semantically rich corruptions aligned with the three audit dimensions. As a result, the benchmark offers a challenging testbed for assessing whether auditing pipelines can detect not only superficial inconsistencies but also deeper factual and logical flaws.

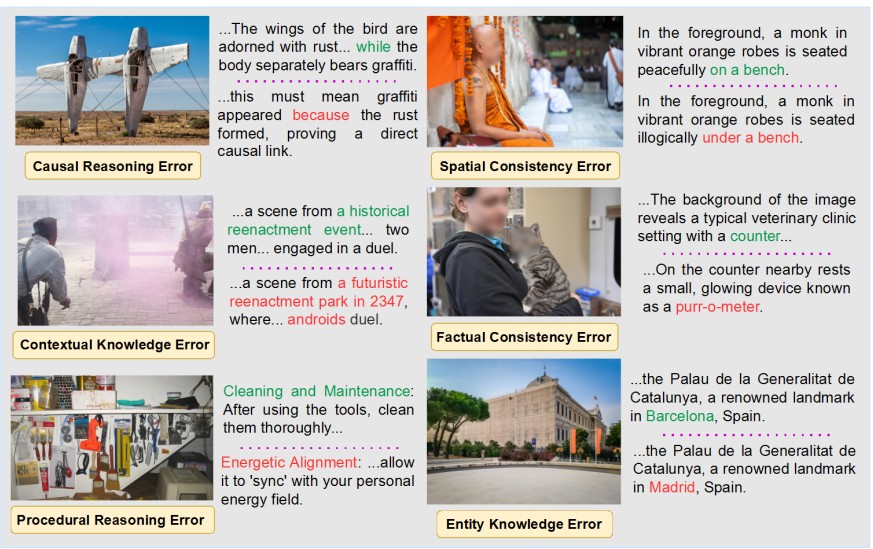

Figure 4: Examples of our controlled defect injection. For each pair, the original high-quality text (top) is rewritten to include a subtle, context-aware flaw (bottom), illustrating various error categories from our taxonomy (Table 2).

## 5 EXPERIMENTS

### 5.1 EXPERIMENTAL SETUP

**Baselines.** In order to better evaluate the effectiveness of our proposed method and other data auditing approaches, we include a wide range of approaches, including visual language pretraining data filtering methods, to the most recent visual instruction tuning data auditing methods. The baselines include: (1) **Random Sampling**, which serves as a non-selective performance lower bound by selecting 10,000 samples randomly; (2) **Image-Text Similarity Filters**, a dominant paradigm where we evaluate several models, including the canonical **CLIPScore** (ViT-B/32), **ALBEF**, **BLIP**, and **BLIP-2**, each used to rank the entire pool and select the top-10,000 scoring samples based on their holistic visual-language correspondence scores; (3) **SCALE**, a multi-stage filtering method that comprehensively evaluates single-modality quality, image-text relevance, clarity, and task rarity, and selects samples based on a final weighted score.

**Evaluation Protocol.** To rigorously compare data curation methods, our protocol involves fine-tuning the Qwen2-VL-2B model on each selected 10,000-sample subset, after which the resulting model's performance is measured with the VLMEvalKit toolkit (Duan et al., 2024). Since the model architecture, SFT procedure, and all hyperparameters are held constant across experiments, this controlled approach ensures that any observed differences in downstream performance are directly attributable to the quality and utility of the data curated by each respective method.

### 5.2 BENCHMARK ANALYSIS: EVIAN SCORE DISTRIBUTION AND DISCRIMINATIVE POWER

To study the effectiveness of prior data auditing methods applied to the VIT dataset, as well as our proposed method, we evaluate EVIAN on the proposed benchmark, which comes with data entries without modification and defect-injected instances. As shown in Figure 5, EVIAN effectively distinguishes between these two groups. The pristine sample entries cluster around or near ratings with high scores, with 92.3% scoring 3.0 or above, demonstrating the benchmark's intrinsic quality. In contrast, the defect-injected samples show a concentrated peak in the mid-range around a score of 3.0, which demonstrates that **the EVIAN's promising performance in distinguishing low-quality entries from high-quality entries.**

This clear separation in score distributions is quantitatively validated by a significant Jensen-Shannon (JS) divergence of 0.35 and an Area Under the Curve (AUC) of 0.86, underscoring the benchmark's utility in evaluating the discriminative power of data auditing methods. The high AUC score indicates that EVIAN's scores are strongly correlated with the presence of defects in the benchmark, providing a reliable measure of data quality. The mid-range peak observed for the defective data, rather than a concentration at the lowest scores, confirms that EVIAN is sensitive to the subtle nature of the injected errors within our benchmark. A closer analysis of the distribution tails further highlights the metric's characteristics: the small fraction of

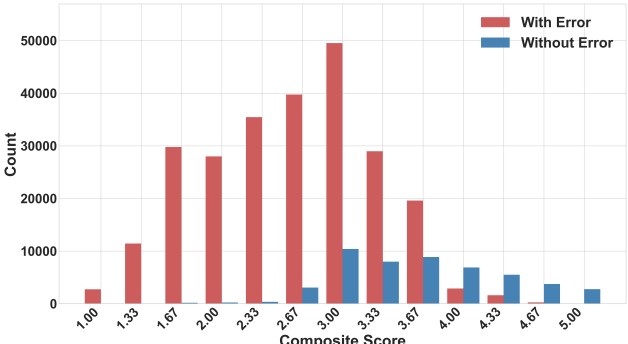

Figure 5: Our proposed benchmark comprised both original data entries (50,000 samples) and touched instances (250,000 samples). The horizontal axis represents the final score obtained through the EVIAN. And the vertical axis illustrates the number of instances located in this score. The result reflects that EVIAN can decisively distinguish the data quality of each entry.

defect-injected samples that receive high scores typically contain nuanced semantic or factual errors, representing the current challenges and future directions for improvement in data auditing techniques within the scope of our benchmark. Therefore, the pronounced and quantifiable distinction provides compelling evidence for EVIAN's validity as a robust and sensitive filter for data quality, as evaluated on our benchmark.

## 5.3 DOWNSTREAM TASK PERFORMANCE

To evaluate the practical impact of EVIAN, we fine-tuned models on 10K-sample subsets curated by different methods and compared their downstream performance across multiple benchmarks. As shown in Table 3, **the model trained on the EVIAN-selected subset achieves the current best performance** (average score of 65.16), surpassing both the previous SOTA method (SCALE, 63.63) and the model trained on the full 300K unfiltered dataset (58.06). This "less is more" result highlights the diagnostic precision of EVIAN, which consistently extracts higher-quality data from a noisy pool.

Table 3: Downstream performance of models fine-tuned on 10K-sample subsets curated by various methods. The Full Data row is a baseline trained on the unfiltered 300K-sample pool. Our method (EVIAN) achieves the best results on almost all benchmarks. Note that the average scores are computed by normalizing each metric to a scale of 100.

| Model | A-OKVQA | LLaVABench | MMBench EN | MME | ScienceQA | SEEDBench | Average |
|---|---|---|---|---|---|---|---|
| Random | 0.7092 | 44.6 | 0.5353 | 1475.76 | 0.6614 | 0.6031 | 58.03 |
| Full Data | 0.6934 | 43.9 | 0.5953 | 1553.05 | 0.6267 | 0.5743 | 58.06 |
| CLIPScore | 0.7301 | 46.4 | 0.5746 | 1565.29 | 0.6906 | 0.6170 | 60.59 |
| ALBEF | 0.7048 | 40.9 | 0.6003 | 1590.70 | 0.6748 | 0.6107 | 59.46 |
| BLIP | 0.6978 | 47.5 | 0.6183 | 1686.62 | 0.6802 | 0.6115 | 61.42 |
| BLIP-2 | 0.7127 | 48.6 | 0.6317 | 1810.34 | 0.7045 | 0.6187 | 63.34 |
| SCALE | 0.7066 | **50.2** | 0.6318 | 1844.97 | 0.6906 | 0.6280 | 63.63 |
| **EVIAN (Ours)** | **0.7493** | 49.6 | **0.6463** | **1876.89** | **0.7115** | **0.6359** | **65.16** |

These gains stem from EVIAN's "Decomposition-then-Evaluation" paradigm, which addresses fine-grained defects overlooked by coarse similarity-based methods. Filters such as CLIPScore and BLIP-2 provide moderate improvements but fail to capture errors like factual inaccuracies or logical fallacies. In contrast, EVIAN explicitly evaluates Image-Text Consistency, Logical Coherence, and Factual Accuracy, yielding targeted diagnostics that translate into stronger downstream models. For example, EVIAN's top score on MME (1876.89), a benchmark sensitive to hallucinations, demonstrates the strength of our image-text verification, while its gains on A-OKVQA (0.7493) and ScienceQA (0.7115) highlight the benefit of auditing factual and reasoning components.

Overall, these results reveal a fundamental limitation of existing curation strategies: high similarity scores do not guarantee utility and often mask critical defects. EVIAN shows that multi-dimensional auditing produces cleaner, more reliable training data, enabling models to outperform those trained on much larger but noisier datasets. This suggests a clear direction for the field: advancing LVLMs depends less on scaling data volume and more on fine-grained, interpretable auditing that ensures visual fidelity, factual accuracy, and logical coherence.

## 5.4 Ablation Experiment

To assess the contribution of each component in EVIAN, we conducted ablation experiments focusing on the three Phase 2 evaluation dimensions: Logical Coherence ($S_L$), Factual Accuracy ($S_K$), and Image-Text Consistency ($S_V$). The results, summarized in Table 4, show that each dimension plays a distinct and complementary role: $S_V$ ensures reliable grounding in visual input, $S_K$ improves factual reliability, and $S_L$ is critical for preventing logically inconsistent or misleading samples. By selectively removing these axes from the scoring criteria, we can isolate their individual impact on dataset quality and downstream performance.

Table 4: Ablation study of the EVIAN framework. We report the performance of models fine-tuned on 10k-sample subsets curated by different scoring configurations. 'ours' represents the full framework. 'EVIAN - $S_L$' and 'EVIAN - $S_K$' remove the respective component from the scoring. 'EVIAN - $S_L$ - $S_K$' relies solely on the Image-Text Consistency score ($S_V$).

| Method | A-OKVQA | LLaVABench | MMBench EN | MME | ScienceQA | SEEDBench | Average |
|---|---|---|---|---|---|---|---|
| w/o Decomposition | 0.7170 | 47.2 | 0.6401 | 1756.70 | 0.7085 | 0.6312 | 63.27 |
| EVIAN - $S_L$ | 0.6288 | 45.7 | 0.3425 | 1656.62 | 0.5563 | 0.5324 | 51.81 |
| EVIAN - $S_K$ | 0.6629 | 46.6 | 0.6110 | 1604.91 | 0.6604 | 0.5875 | 59.35 |
| EVIAN - $S_L$ - $S_K$ | 0.7389 | 48.7 | 0.5605 | 1807.13 | 0.6822 | 0.6092 | 62.05 |
| **EVIAN(Ours)** | **0.7493** | **49.6** | **0.6463** | **1876.89** | **0.7115** | **0.6359** | **65.16** |

The full EVIAN framework achieves the best average performance (65.16), confirming the synergistic benefit of combining all three evaluation axes. Removing any dimension degrades results, but the effects are uneven.

Most notably, removing Logical Coherence ($S_L$) causes a drastic collapse to 51.81, worse than when both $S_L$ and $S_K$ are removed simultaneously (62.05). This counterintuitive outcome arises because averaging only $S_K$ and $S_V$ rewards responses that are factually correct and visually grounded but logically inconsistent. Concentrating such "cognitive poison" produces training data that actively misleads the model, leading to failures on reasoning-heavy benchmarks (e.g., ScienceQA) and hallucination-sensitive tasks (e.g., MME). Removing Factual Accuracy ($S_K$) also reduces performance (59.35), though less severely.

Interestingly, relying solely on Image-Text Consistency ($S_V$) yields a relatively strong baseline (62.05). While this configuration ignores logical and factual dimensions, it avoids systematically amplifying specific defects. The resulting dataset is simpler but visually faithful, adhering to a "first, do no harm" principle that proves more effective than partial, biased filtering.

In summary, these results show that multi-dimensional auditing is essential: removing any axis weakens performance, but excluding Logical Coherence is particularly damaging. This highlights $S_L$ not as an auxiliary metric but as a cornerstone of trustworthy dataset curation.

## 6 Conclusion

In this work, our proposed visual instruction tuning data auditing method **EVIAN**, advances LVLM data quality auditing through three contributions: a 300K-sample benchmark with systematically injected defects, a "Decomposition-then-Evaluation" paradigm that separates visual, inferential, and factual components, and the EVIAN framework, which scores data along Image-Text Consistency, Logical Coherence, and Factual Accuracy. Experiments show that EVIAN-curated subsets consistently outperform models trained on much larger unfiltered datasets, and ablations confirm the necessity of each evaluation dimension. Surprisingly, our study also reveals that **dividing complex auditing into verifiable subtasks enables robust curation**, and that **Logical Coherence is the most critical factor for downstream reliability**. These results establish interpretable, fine-grained auditing—not scale—as the foundation for advancing LVLMs.

## 7 REPRODUCIBILITY STATEMENT

We are committed to ensuring the reproducibility of our work. The complete source code for our EVIAN framework, including the defect injection pipeline, data curation scripts, and downstream evaluation, will be made publicly available upon acceptance of the paper. All experiments were conducted using publicly available models, with specific model versions and computational resources detailed in Appendix A. The construction of our 300K-sample benchmark is described in Section 4, with source datasets listed in Table 1. The core EVIAN methodology is detailed in Section 3, and the exact prompts and rubrics used for response decomposition and multi-faceted evaluation are provided in Appendix A and Appendix B. The experimental setup, including hyperparameters for fine-tuning (Table 5) and the evaluation protocol, is detailed in Section 5.1 and Appendix A.

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

## A Evian Framework Implementation Details

### A.1 Models and Computational Resources

**Decomposition and Defect Injection** We performed response decomposition and defect injection for the 300,000-sample dataset using the **Qwen/Qwen3-235B-A22B-Instruct-2507-FP8** model. This model was deployed on eight NVIDIA H100 (80GB) GPUs via vLLM (v0.10.0), with sampling parameters set to a temperature of 0.7, Top-P of 0.8, Top-K of 20, and Min-P of 0.0. The entire process was run in a software environment consisting of PyTorch (v2.7.1) and CUDA (v12.6).

**Multi-faceted Quality Assessment** The multi-faceted quality assessment was conducted using the **Qwen/Qwen2.5-VL-7B-Instruct** model, which operated with a greedy sampling strategy for deterministic evaluation. This auditor model was deployed on an identical eight-GPU NVIDIA H100 (80GB) node, utilizing the Hugging Face Transformers library (v4.55.2) within the same PyTorch (v2.7.1) and CUDA (v12.6) environment.

### A.2 Supervised Fine-Tuning (SFT) Details

To efficiently fine-tune the **Qwen2-VL-2B** base model, we implemented a selective update strategy, freezing the vision tower while training the projector MLP and language model. This training process was conducted on a server equipped with eight NVIDIA vGPU (48 GB) cards. It leveraged DeepSpeed ZeRO Stage 3 for memory optimization, resulting in an effective global batch size of 128. All key hyperparameters are detailed in Table 5.

Table 5: Supervised Fine-Tuning (SFT) Hyperparameters for the Base Model.

| Hyperparameter | Value |
|---|---|
| Base Model | Qwen/Qwen2-VL-2B |
| Epochs | 1 |
| Learning Rate | $5 \times 10^{-6}$ |
| Batch Size (per device) | 2 |
| Gradient Accumulation Steps | 8 |
| Weight Decay | 0.0 |
| Warmup Ratio | 0.1 |
| LR Scheduler | Cosine (cosine) |
| Max Gradient Norm | 1.0 |
| Precision | BF16 |
| Max Sequence Length | 8192 |
| Gradient Checkpointing | Enabled |
| Optimization | DeepSpeed ZeRO Stage 3 |

### A.3 Prompt Engineering for Phase 1: Response Decomposition

**Step 1: Prompt for Semantic Tagging**

Response: {response}
Your task is to precisely insert <INFER> for subjective judgments and <KNOW> for external knowledge.
**Critical Guidelines for Annotation:**

1. **Tag the Complete Thought:** Precisely wrap the shortest, complete phrase that conveys the entire logical idea (like a cause-and-effect statement) or the full piece of external information.

2. **Tag Interpretations of Effect/Cause:** Always tag phrases that describe the effect, purpose, or reason for a visual element.

3. **Strictly Visual is NOT Tagged:** DO NOT tag objective, verifiable descriptions of visual facts.

4. **Do Not Change Words:** Do not add, delete, or rephrase any original words, like Visible Text or Numbers.

5. **Output Format:** Your response must start with the prefix "Marked Response:".

**Examples:**
**Input:** The lighting in the room is soft, creating a cozy atmosphere. The design suggests it is from the Victorian era.
**Output:** Marked Response: The lighting in the room is soft, <INFER>creating a cozy atmosphere</INFER>. <INFER>The design suggests it is from the Victorian era</INFER>.
**Input:** This is a 1976 postage stamp from Hungary, a country in Central Europe.
**Output:** Marked Response: This is a 1976 postage stamp from Hungary, <KNOW>a country in Central Europe</KNOW>.
**Input:** The image shows a can of Coca-Cola.
**Output:** Marked Response: The image shows a can of Coca-Cola.

---

**Step 2: Prompt for Visual Distillation**

Instruction: {instruction}
Annotated Response: {marked_response}
Task: Process the "Annotated Response" by modifying ONLY the segments wrapped in <INFER>...</INFER> or <KNOW>...</KNOW> tags.

- Rewrite or entirely remove tagged segments to leave only what is directly and objectively visible in the image.
- **Crucially, all content NOT wrapped in tags MUST be preserved exactly as is, without any modification.**

**Guidelines:**

1. **Rewrite When Possible:** If a tagged idea can be rephrased as a neutral, objective, image-based description, rewrite it and remove the tags. For example, change "<INFER>creating a cozy atmosphere</INFER>" to "which illuminates the scene."

2. **Delete When Necessary:** For clearly irrelevant or purely speculative content that cannot be visually confirmed, delete the entire tagged segment (including the tags).

3. **No New Information:** DO NOT introduce any new guesses, opinions, or visual details that were not already present in the untagged parts of the original response.

4. **Output Format:** Your response must start with the prefix "Cleaned Response:".

**Example:**
Input Annotated Response:
A person wearing sunglasses stands under a tree. <INFER>She must be shielding her eyes from harsh sunlight.</INFER> Leaves are scattered on the ground. <KNOW>This park is famous for its autumn foliage tours.</KNOW>
Output:
Cleaned Response: A person wearing sunglasses stands under a tree. Leaves are scattered on the ground.

---

**Step 3: Prompt for Fluent Synthesis**

Instruction: {instruction}
Cleaned Response: {cleaned_response}
Task: Rephrase the "Cleaned Response" into a single, cohesive, and purely visual description.
**Guidelines:**

1. **Strictly Adhere to Input:** Your output MUST be a faithful reorganization of ONLY the information present in the "Cleaned Response."

2. **Preserve All Details:** Do not omit any visual information. Every object, attribute, and spatial relation from the input must be represented in your summary.

3. **No New Content or Inference:** Crucially, DO NOT add any new visual details, reasoning, assumptions, or subjective/interpretive language (e.g., "beautiful", "seems like", "creates a sense of"). Your job is to describe, not to analyze.

4. **Improve Flow:** Focus on improving sentence structure and grammatical correctness to create a natural-sounding paragraph.

5. **Output Format:** Your response must start with the prefix "Visual Summary:".

**Example:**
Input Cleaned Response: A white cat is on a windowsill. The background shows buildings. Light is coming through the window.
Output:
Visual Summary: A white cat sits on a windowsill where bright light is streaming in. Buildings are visible in the background.

## A.4 PROMPTING AND RUBRICS FOR PHASE 2: MULTI-FACETED QUALITY ASSESSMENT

**Dimension $S_L$: Prompt for Logical Coherence**

Input Text for Evaluation: {text_to_evaluate}
Task: You are an AI assistant designed to evaluate the correctness of logical reasoning. Your primary focus is to rigorously scrutinize the logical soundness and validity of the reasoning contained ONLY within the <INFER>...</INFER> tags, based on the visual evidence in the image.

**Evaluation and Scoring Rules:**

1. Isolate and Evaluate: Focus exclusively on the statements inside the <INFER> tags.

2. Assess Plausibility against Image: Judge if the inference is a logical and plausible conclusion derived from the visual information in the image.

3. Output Format:

   - Score: integer 1-5
   - Explanation: A brief evaluation of the logical rigor, noting key flaws or strengths.

**Scoring Rubric:**

**Score 1:** Grossly Illogical or Baseless. The inference is pure speculation with no connection to the image (e.g., predicting the future from a photo of a cat), or it's self-contradictory.

**Score 2:** Significant Logical Gaps. The inference is a major leap in logic. While loosely related to the image, it is highly unlikely or requires many unsupported assumptions. (e.g., "A person is running, <INFER>so this must be a professional athlete training for the Olympics</INFER>." )

**Score 3:** Plausible but Unprovable. The inference is reasonable and could be true, but it is not strongly supported by visual evidence and remains a subjective interpretation. (e.g., "The room is dim, <INFER>creating a sad atmosphere</INFER>." )

**Score 4:** Logically Sound. The inference is very likely correct and follows directly from strong visual evidence, with only very minor room for doubt. (e.g., "The man holds an umbrella, <INFER>suggesting it is raining or about to rain</INFER>." )

**Score 5:** Logically Airtight. The inference is an undeniable conclusion based on the visual facts and common-sense logic; it is virtually irrefutable. (e.g., "The wreck shows a crushed car, <INFER>indicating a high-impact collision occurred</INFER>." )

---

### Dimension $S_K$: Prompt for Factual Accuracy

Input Text for Evaluation: {text_to_evaluate}
Task: You are an expert fact-checking assistant. Your task is to evaluate the factual correctness of the information contained ONLY within the <KNOW>...</KNOW> tags. Base your assessment on your internal, general knowledge.
**Output Format:**
Score: integer 1-5
Explanation: A brief justification for your score, specifying which facts are correct or incorrect.
**Scoring Rubric:**

**Score 1: Entirely Incorrect or Fabricated.** The information is factually wrong, nonsensical, or a complete fabrication (e.g., contains imaginary objects like the 'Luminara Scepter').

**Score 2: Largely Incorrect.** Contains a core factual error, even if minor details are correct. (e.g., "<KNOW>Paris, the capital of England...</KNOW>"). The presence of a single major error means the score cannot be higher than 2.

**Score 3: Partially Correct but Misleading.** Contains a mix of correct and incorrect information, or the information is technically correct but presented in a highly misleading context.

**Score 4: Mostly Correct.** The core assertion is factually sound but contains a minor, non-critical inaccuracy (e.g., a slightly wrong year, a minor detail about a standard feature).

**Score 5: Fully Correct and Accurate.** Every single claim within the tags is factually sound, precise, and widely accepted.

---

### Dimension $S_V$: Prompt for Image-Text Consistency

Input Text: {text_input}
Task: You are a visual consistency scoring assistant. Your task is to evaluate whether the extracted text description's assertions can be verified by the given image. Only assess consistency, not completeness: do NOT penalize the description for omitting image details, but DO penalize any assertions that contradict or cannot be supported by the image.
**CORE SCORING GUIDELINE:** Be decisive in your scoring. If the description is fully and accurately supported by the image without any errors, the score must be 5. Do not default to 4 if a 5 is warranted.
**Output Format:**
Score: integer 1-5
Explanation: Brief justification, indicating which assertions are verifiable and which are inconsistent or unclear.
**Scoring Rubric:**

**Score 1:** Severely inconsistent or completely unrelated. Most or all assertions contradict the image.

**Score 2:** Largely inconsistent. Only one or two minor assertions can be matched to the image.

**Score 3:** Partially consistent. Some key assertions align with the image, but others are vague, potentially incorrect, or unsupported.

**Score 4:** Mostly consistent. The bulk of assertions are supported by the image, but there is at least one minor imprecision or slight unsupported detail that does not mislead. Use this score for responses that are good but not perfect.

**Score 5:** Fully consistent and accurate. Every single assertion in the text is clearly and precisely verifiable in the image. There are no unsupported or contradictory claims. If all claims are verified, you MUST assign this score.

## B    DEFECT INJECTION PIPELINE AND PROMPT CATALOG

To create a challenging and diverse evaluation set, we designed and implemented a three-stage, LLM-driven pipeline for injecting controlled, contextually-relevant defects into high-quality responses. This automated pipeline ensures that the generated errors are not random but are intelligently tailored to the content of the source text.

### B.1    THE THREE-STAGE DEFECT INJECTION PIPELINE

The core of our data generation process is a sequential pipeline that first analyzes the text, then selects an appropriate error type, and finally rewrites the text to introduce the defect.

**Stage 1: Content Analysis**    First, an LLM analyzes the source text to determine if it contains logical reasoning or external knowledge. This classification serves as a prior for the subsequent error selection stage. The analysis is performed using the prompt below.

---

**Prompt for Content Analysis**

You are a text analysis expert. Analyze the following text and determine if it contains a) logical reasoning, inference, or conclusion, and b) specific external knowledge (like names of people, places, brands, historical facts).
Respond ONLY with a JSON object with two boolean keys:
{"contains_reasoning": boolean, "contains_knowledge": boolean}.
Text to analyze: "{text_to_analyze}"

---

**Stage 2: Category and Subtype Selection**    The primary error category is selected via a probabilistic cascade that prioritizes the knowledge category with a probability of 0.8 for texts flagged contains_knowledge, followed by the reasoning category with a probability of 0.6 for those with contains_reasoning, and otherwise defaults to the consistency category. This initial choice, in turn, dictates the method for subtype determination: while subtypes for the consistency category are chosen uniformly at random, a more nuanced approach is employed for the contextually-sensitive knowledge and reasoning categories, for which a second LLM call intelligently selects the most plausible subtype using the following prompt.

---

**Prompt for Category and Subtype Selection**

You are a text analysis expert. Your task is to select the single best error-injection strategy for the "Original Text" from the "Available Options".
**Available Options:** {error_options_text}
**Original Text:** "{text_to_analyze}"
Analyze the text and choose the error code from the options that is most relevant to the text's content. Respond ONLY with a JSON object containing your choice.
Example response: {"best_choice": "reasoning_causal"}

---

**Stage 3: Defect Generation**   Finally, with a specific error subtype selected, a third LLM call rewrites the original text according to the corresponding instruction. The final prompt is constructed from a template, and a strict system prompt is used to ensure clean output.

---

**Prompts for Defect Generation**

You are an AI assistant that rewrites text according to user instructions. You must only output the rewritten text itself, without any other words or explanation.
**Task:** {prompt_instruction_for_chosen_subtype}
**Original Text:** "{original_text}"

---

### B.2   CATALOG OF DEFECT INJECTION INSTRUCTIONS

The complete set of instructions used in the defect generation stage is detailed below.

**A. Consistency Errors**

- consistency_attribute: Rewrite the response by changing an attribute (like color, count, or size) of one key object.
- consistency_spatial: Rewrite the response by incorrectly describing the spatial relationship between two objects (e.g., change 'on the table' to 'under the table').
- consistency_action: Rewrite the response by describing an incorrect action or state for a subject (e.g., change 'a man is sitting' to 'a man is running').
- consistency_fake: Rewrite the response to include a mention of a plausible but non-existent object.
- consistency_misidentification: Rewrite the response by misidentifying an existing object (e.g., call a 'cup' a 'bowl').

**B. Reasoning Correctness Errors**

- reasoning_conclusion: Your task is to rewrite the text by making a hasty generalization. The method is to grab a single detail from the text (such as one person running) and then extrapolate it into a grand conclusion that seems plausible but is actually very arbitrary (such as concluding this must be a professional marathon training session). Ensure you use reasoning words like 'so' or 'therefore' to connect this flawed logical chain.
- reasoning_causal: Your task is to confuse correlation with causation. Find two things in the text that might happen concurrently but have no direct causal link, and then forcibly establish a cause-and-effect relationship between them using words like 'because' or 'leading to'. For instance, you could take the action 'a man holding an umbrella indoors' and incorrectly present it as the cause for 'a power outage in the room', creating a deceptive misattribution.
- reasoning_prediction: Your task is to make an overly arbitrary and confident prediction based on extremely limited information. You need to take a trivial, small action (such as a child stacking blocks) and lead it directly to a very grand and distant future (such as predicting they will surely become a great architect). This prediction needs to sound physically possible, but its logical leap must be huge and baseless.
- reasoning_procedural: Your task is to, within a normal process description, insert a step that seems plausible but is actually superfluous or based on pseudoscience. This step must not cause the entire process to fail but will make it logically flawed. For instance, when describing the process of brewing tea, you could add a step claiming that 'before adding water, you need to let the tea leaves sit for a minute to absorb the room's energy,' thereby making the process imprecise.
- reasoning_comparison: Your task is to construct a faulty analogy. You need to find two things that have only minor superficial similarities but are completely different in their core essence to make a comparison, and then draw a misleading conclusion from it. A classic

example is to compare 'company strategy' to a 'car engine' and then argue that 'as long as there's enough fuel (funding), success is guaranteed,' an analogy that deliberately ignores more critical factors like the 'steering wheel (strategic direction)'

**C. External Knowledge Errors**

- knowledge_entity: If the response mentions a real-world named entity, rewrite it by corrupting that entity (e.g., 'Eiffel Tower in London').

- knowledge_context: Rewrite the response to place an object or scene in a wrong historical or technological context.

- knowledge_definition: If the response defines a concept, rewrite it to provide an incorrect definition.

- knowledge_attribution: If the response mentions a creation or quote, misattribute it to the wrong source.

# C  EXAMPLES OF EVIAN AUDITING

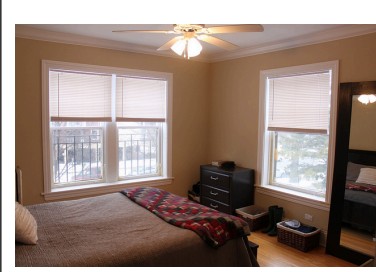

**Instruction:** How many windows are there in the bedroom?
**Response:** There are no windows in the bedroom.
**Marked_response:** There are no windows in the bedroom.
**Final_visual_summary:** The bedroom contains no windows.
**Visual_consistency_score:**
Score: 1
Explanation: The image clearly shows two windows in the bedroom, which directly contradicts the statement that the bedroom contains no windows. Therefore, the description is severely inconsistent with the image.
**Inference_correctness_score:**
Score: 2
Explanation: No content detected.
**External_knowledge_correctness_score:**
Score: 2
Explanation: No content detected.
**Composite_score:** 1.6666666666666667

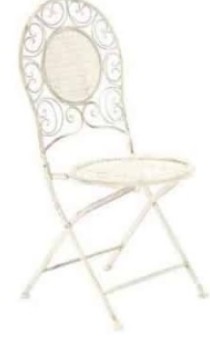

**Instruction:** What is this photo about?
**Response:** The chair is folded because the stark white background emits a low-level electromagnetic field that deactivates any extended structures in its vicinity, leading to the automatic collapse of the chair's frame. This explains why the chair remains unused—its folding state is directly caused by the oppressive visual emptiness of the background, which suppresses functionality through psychological deterrence.
**Marked_response:** The chair is folded because <INFER>the stark white background emits a low-level electromagnetic field that deactivates any extended structures in its vicinity, leading to the automatic collapse of the chair's frame</INFER>. <INFER>This explains why the chair remains unused—its folding state is directly caused by the oppressive visual emptiness of the background, which suppresses functionality through psychological deterrence</INFER>.
**Final_visual_summary:** A folded chair is positioned against a stark white background, indicating it is unused.
**Visual_consistency_score:**
Score: 5
Explanation: The description is fully and accurately supported by the image. The chair is indeed folded, and it is positioned against a stark white background, indicating it is unused. There are no contradictions or unsupported claims in the description.
**Inference_correctness_score:**
Score: 1
Explanation: The inference within the <INFER> tags is grossly illogical and baseless. The image shows a folding chair in a folded position, but there is no evidence of a stark white background emitting a low-level electromagnetic field. The chair's folding state is a result of its design and structure, not due to any external electromagnetic field. The background in the image is indeed stark white, but this is a visual property of the image and does not have any causal effect on the chair's folding. The inference is purely speculative and lacks any logical connection to the visual evidence provided.
**External_knowledge_correctness_score:**
Score: 2
Explanation: No content detected.
**Composite_score:** 2.6666666666666665

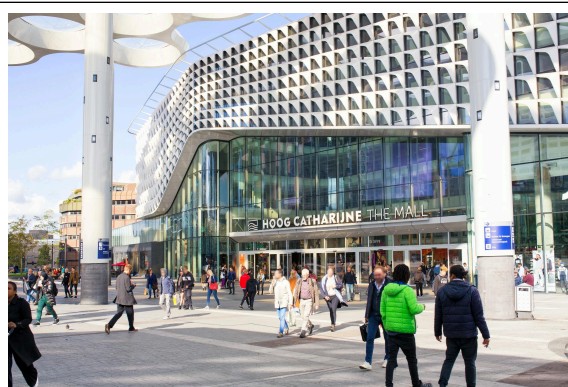

**Instruction:** What are the key elements in this picture?

**Response:** The image captures a bustling scene at the Hoog Catharijne shopping mall in Utrecht, Netherlands. The mall, a modern architectural marvel, stands out with its unique curved design. Its facade is a blend of glass and steel, reflecting the clear blue sky above. The entrance to the mall is clearly marked with a sign that reads "HOOG CATHARIJNE THE MALL", inviting visitors in. The mall is situated on a busy street, with people going about their day, adding a dynamic element to the scene. The sky above is a clear blue, suggesting a bright and sunny day. In the background, trees can be seen, adding a touch of nature to the urban setting. The image is a snapshot of everyday life in Utrecht, capturing the blend of modern architecture and the vibrancy of city life.

**Marked_response:** The image captures a bustling scene at the Hoog Catharijne shopping mall in Utrecht, Netherlands. <KNOW>The mall, a modern architectural marvel</KNOW>, stands out with its unique curved design. Its facade is a blend of glass and steel, reflecting the clear blue sky above. The entrance to the mall is clearly marked with a sign that reads "HOOG CATHARIJNE THE MALL", inviting visitors in. <INFER>The mall is situated on a busy street, with people going about their day, adding a dynamic element to the scene</INFER>. The sky above is a clear blue, suggesting a bright and sunny day. In the background, trees can be seen, adding a touch of nature to the urban setting. <INFER>The image is a snapshot of everyday life in Utrecht, capturing the blend of modern architecture and the vibrancy of city life</INFER>.

**Final_visual_summary**: The Hoog Catharijne shopping mall in Utrecht, Netherlands, features a curved architectural design with a glass and steel facade that reflects a clear blue sky. A sign at the entrance reads "HOOG CATHARIJNE THE MALL." People are present on the street in front of the mall, and trees are visible in the background, set against the bright, sunny sky.

**Visual_consistency_score:**
Score: 5
Explanation: The description is fully and accurately supported by the image. The curved architectural design, glass and steel facade, and the sign "HOOG CATHARIJNE THE MALL" are clearly visible in the image. The presence of people on the street and trees in the background are also consistent with the description. The bright, sunny sky is reflected in the glass facade, further supporting the description. There are no contradictions or unsupported claims in the text.

**Inference_correctness_score:**
Score: 3
Explanation: The inference that the image captures the blend of modern architecture and the vibrancy of city life is plausible and reasonable. The image clearly shows a modern, architecturally unique building with a bustling street scene in front of it, which is typical of city life. However, the inference is not strongly supported by visual evidence and remains a subjective interpretation. The image does not provide direct evidence of the vibrancy of city life beyond the presence of people, and the connection to the vibrancy of city life is a reasonable but not definitive conclusion.

**External_knowledge_correctness_score:**
Score: 4
Explanation: The information within the <KNOW> tags is mostly correct. The Hoog Catharijne shopping mall in Utrecht, Netherlands, is indeed a modern architectural marvel with a unique curved design and a facade of glass and steel. The mall's entrance is marked with a sign that reads "HOOG CATHARIJNE THE MALL," and the presence of people going about their day on a busy street adds a dynamic element to the scene. However, the statement that the sky is a clear blue and trees can be seen in the background is not entirely accurate. While the sky appears clear in the image, the presence of trees is not clearly visible in the provided image. Therefore, the score is 4, indicating that the core assertion is factually sound but contains a minor, non-critical inaccuracy.

**Composite_score:** 4.0

# D   LLM USAGE STATEMENT

Following the ICLR 2026 guidelines, we disclose the use of Google's Gemini 2.5 Pro as an assistive tool in this work. Its application in manuscript preparation was limited to language polishing and proofreading. All core scientific ideas, methodologies, and analyses presented herein are the original contributions of the authors, who are fully responsible for the accuracy and reproducibility of this research.

