# OpenReview forum: "Evian: Towards Explainable Visual Instruction-tuning Data Auditing"
_ICLR.cc/2026/Conference — ICLR 2026 Conference Withdrawn Submission_

### Official Review · Reviewer_oEZv · 2025-10-28

**Soundness:** 3
**Presentation:** 2
**Contribution:** 3
**Rating:** 4
**Confidence:** 4

**Summary:**

The paper addresses the data quality bottleneck in training Large Vision-Language Models by proposing fine-grained auditing instead of coarse filtering. It builds a 300K-sample benchmark with systematically injected, subtle defects, and introduces a Decomposition-then-Evaluation paradigm that separates responses into visual description, subjective inference, and factual claim. The EVIAN pipeline evaluates these components along Image-Text Consistency, Logical Coherence, and Factual Accuracy. Experiments show that models fine-tuned on EVIAN-curated, compact, high-quality data outperform those trained on much larger datasets, with Logical Coherence emerging as the most impactful quality dimension.

**Strengths:**

1. The problem this paper studies is a critical problem in visual instruction tuning.
2. The proposed auditing method is technically sound and performs well as demonstrated by experiments.
3. The formulated multi-faceted quality assessment is also practical and reasonable.

**Weaknesses:**

1. Limited baselines for comparisons. Since a well-trained MLLM (Qwen2.5-VL) is involved in the proposed method, some important baselines are available. For example, what about distilling from this MLLM to clean the dataset. In addition, can we audit the datasets by directly prompting the MLLM?)
2. Lack of **direct** comparisons on the performance of different auditing methods. Although Table 3 shows the proposed method is superior to existing ones in terms of MLLM benchmarks scores, this is **indirect**. Can the authors compare the auditing performance of the baselines (existing ones and the direct prompting method mentioned in weakness 2) in a **direct** manner? The authors could plot a precision-recall curve for all the compared methods.
3. Lack of results on per-category dataset audition. Can the authors report the auditing performance (direct evaluation) for each category studied (e.g., categories in Figure 4)?

**Questions:**

1. What models are involved in the SCALE baseline?
2. The authors use Qwen2.5-VL-7B to perform audition. However, this model might perform poorly for tasks like fact checking or logical reasoning. The authors are suggested to use its larger version, e.g., Qwen2.5-VL-72B.

---

### Official Review · Reviewer_18vL · 2025-10-29

**Soundness:** 2
**Presentation:** 1
**Contribution:** 2
**Rating:** 2
**Confidence:** 5

**Summary:**

This paper introduces EVIAN, a framework for auditing visual instruction-tuning data. The framework decomposes model responses into visual, inferential, and factual components, which are subsequently evaluated for consistency, logical soundness, and factual accuracy. Furthermore, the paper presents a 300K-sample benchmark containing systematically injected defects to assess the data auditing capability of the proposed pipeline. Experimental results on this benchmark demonstrate that models trained on EVIAN-curated data consistently outperform several baselines.

**Strengths:**

This paper investigates the evaluation, cleaning, and construction of high-quality visual instruction tuning datasets—a critical problem for advancing VLMs.

**Weaknesses:**

1. The core approach relies on using LLMs to decompose and re-annotate existing annotations. However, compared to prior LLM/VLM-based data-cleaning works [1–4], this paper lacks significant technical novelty. The primary distinction lies in the specific text prompts employed. Additionally, the paper doesn’t properly discuss related works on LLM- or VLM-assisted data refinement, especially in the context of instruction-tuning [1–4]

    [1] Chen et al. Alpagasus: Training a Better Alpaca with Fewer Data.

    [2] Wei et al. InstructionGPT-4: A 200-Instruction Paradigm for Fine-Tuning MiniGPT-4.

    [3] Wang et al. Finetuned Multimodal Language Models Are High-Quality Image-Text Data Filters.

    [4] Zhang et al. Reflective Instruction Tuning: Mitigating Hallucinations in Large Vision-Language Models.


2. The experiments are all conducted on a custom benchmark that uses synthetic noise. This is problematic because it’s not representative of real-world data. Since the noise is artificially added, models might just learn to pick up superficial patterns rather than solving noise-related issues. As a result, the reported results don’t give us a reliable picture of how the method would work in real-world scenarios.
3. The evaluation omits several critical baselines. Notably, the performance of Qwen2-VL and the results of finetuning baselines on clean data are not reported. In addition, the proposed method uses a 235B-parameter LLM, while most of the compared models are much smaller. This huge gap in model size makes the comparisons unfair.
4. The proposed pipeline is overly complex and computationally expensive. It remains unclear how it would perform relative to a simpler yet strong alternative, e.g., re-annotating data using a state-of-the-art VLM of comparable size.
5. One big issue with noisy instruction-tuning data is that it can make hallucination problems. However, the paper doesn’t include any evaluation on commonly used hallucination benchmarks, such as POPE.
6. The paper have several readability issues:
    - The notations *I*, *P*, and *R* (line 161) are not introduced.
    - Figure 3’s color scheme is confusing and poorly explained—it’s not introduced properly in the figure, caption, or text.

**Questions:**

Please see weakness.

---

### Official Review · Reviewer_TRi1 · 2025-10-30

**Soundness:** 3
**Presentation:** 3
**Contribution:** 3
**Rating:** 4
**Confidence:** 5

**Summary:**

This paper addresses the critical problem of data quality in the training of LVLMs, particularly for visual instruction tuning. The authors argue that existing data filtering methods, which often rely on coarse-grained similarity scores, fail to detect subtle but crucial semantic flaws like logical fallacies, factual inaccuracies, or hallucinations.
To tackle this, the paper introduces three core contributions:

1. A new paradigm called "Decomposition-then-Evaluation," which breaks down complex model responses into three verifiable components: pure visual descriptions, subjective inferences, and external factual claims.

2. An instantiation of this paradigm in a pipeline named EVIAN. EVIAN first decomposes the responses and then evaluates them along three distinct axes: Image-Text Consistency, Logical Coherence, and Factual Accuracy.

3. A large-scale, 300K-sample benchmark specifically designed for auditing, which includes a "gold standard" set and a "challenge" set created by systematically injecting a diverse taxonomy of defects.

The authors' key empirical finding is that a model fine-tuned on a small, 10K-sample subset curated by EVIAN significantly outperforms models trained on the entire 300K unfiltered dataset and on subsets selected by other filtering methods. Furthermore, an ablation study reveals that Logical Coherence is the most critical dimension for data quality, as its exclusion leads to a drastic degradation in downstream performance.

**Strengths:**

1. The quality of training data is arguably one of the most significant bottlenecks for advancing LVLMs. This paper moves beyond simplistic "more data is better" or coarse filtering approaches to tackle the nuanced and critical issue of semantic and logical data integrity

2. The "Decomposition-then-Evaluation" paradigm is a strong conceptual contribution. It is an elegant way to reframe the intractable problem of evaluating a complex, open-ended response into a set of more manageable, verifiable sub-problems. The three evaluation axes are well-chosen and cover the primary failure modes of current LVLMs.

**Weaknesses:**

1.  **Heavy Reliance on a Superior Model as Auditor:** The framework's core logic depends heavily on using large, powerful models as auditors, which raises fundamental questions about the source of the observed improvements. This issue is particularly pronounced in the paper's experimental setup:

    *   **A Form of Knowledge Distillation:** The primary experiments involve fine-tuning a `Qwen2-VL-2B` model on data curated by a more advanced `Qwen2.5-VL-7B` model from the same family. This setup can be interpreted as a form of knowledge distillation, where a stronger "teacher" model provides high-quality signals (the EVIAN scores) to guide the training of a smaller "student" model. Consequently, it becomes difficult to disentangle the benefits of the EVIAN methodology itself from the benefits of simply having a superior model guide a weaker one. The gains might stem more from the raw capability of the teacher model than from the intrinsic value of the decomposition framework.

    *   **The "Sufficiently Powerful Auditor" Paradox:** This leads to a more profound question: if the goal is to improve an already SOTA base model, can we always find a significantly more capable model to serve as a reliable auditor? This approach may work for improving smaller models, but its utility for pushing the frontier of the most advanced models is questionable, as it presupposes the existence of a "super-SOTA" judge.

2.  **Scalability Concerns:** Recent work, such as the analysis from the FineVision paper, has explored data curation at a much larger scale. One of their key findings suggests that once the dataset size reaches a certain critical mass, the negative impact of removing even "low-quality" samples may outweigh the benefits of training on a smaller, cleaner set. The study notes:
    > "This could indicate that with a sufficiently large dataset that you train on long enough, it hurts more to remove samples, even if they were judged to be of low quality, than to train on them."

**Questions:**

Would your conclusion still hold when scaling to models and datasets that are orders of magnitude larger?

---

### Official Review · Reviewer_CEez · 2025-11-03

**Soundness:** 3
**Presentation:** 3
**Contribution:** 2
**Rating:** 4
**Confidence:** 5

**Summary:**

This paper presents EVIAN, a novel explainable data auditing framework for Visual Instruction Tuning in VLMs. The method follows a Decomposition-then-Evaluation paradigm that decomposes model responses into visual, inferential, and factual components and assesses them along three interpretable dimensions—Image-Text Consistency, Logical Coherence, and Factual Accuracy. The authors further construct a 300K-sample benchmark with systematically injected semantic defects to facilitate fine-grained auditing. Empirical results show that models fine-tuned on EVIAN-curated data outperform those trained on much larger unfiltered datasets, suggesting that data quality outweighs data scale in VLMs training.

**Strengths:**

- The Decomposition-then-Evaluation approach offers an interpretable and modular framework for assessing data quality, moving beyond opaque single-score filtering metrics like CLIPScore.
- The authors introduce a 300K-sample benchmark with controlled semantic defect injection—providing a rare, well-structured dataset for studying fine-grained multimodal data auditing.
- EVIAN consistently outperforms previous baselines such as SCALE and CLIP-based methods across multiple downstream benchmarks (e.g., MME, ScienceQA), demonstrating robust generalizability.
-  The paper includes detailed analyses that isolate the contribution of each dimension (SL, SK, SV), clearly showing Logical Coherence as the most critical factor.

**Weaknesses:**

- The auditing pipeline relies heavily on Qwen-series models (e.g., Qwen3-235B and Qwen2.5-VL-7B). This raises concerns about model bias transfer, as the auditor and the evaluated data share similar architectures and training distributions.
- The benchmark’s defect injection procedure—though systematic—depends on LLM-generated corruptions. These artificial errors may not fully capture the complexity or distribution of real-world data flaws. The paper could improve by including evaluations on naturally noisy datasets or human-annotated errors.
- While EVIAN improves data quality, the computational cost of multi-step decomposition and evaluation (each requiring multiple large model calls) is not quantified.
- The final selection score uses an equal-weight averaging scheme (Eq. 2). Although the authors note tunability, they do not show how varying weights across tasks affects performance. A sensitivity analysis would better justify the robustness of this design choice.

**Questions:**

Are there any human evaluation results confirming that high EVIAN scores correspond to perceived high-quality samples?

---

### Author Response · Authors · 2025-11-26
**Response to Reviewer CEez**

We appreciate the opportunity to clarify our design choices with new, detailed experimental results regarding model bias, sensitivity analysis, and computational efficiency.

**1. Model Bias & Human Alignment (W1 & Q1)**

We acknowledge the concern that using Qwen-series models for both auditing and baselines might introduce inductive bias. To address this and rigorously verify alignment with human perception, we conducted a new correlation study.

*   **Experimental Setup:** We randomly sampled **$N=200$** instances from our benchmark, stratified to cover the full quality spectrum.
*   **Method:** We compared the scores assigned by **EVIAN** against two independent evaluators:
    1.  **Human Experts:** Three annotators provided ground-truth quality ratings.
    2.  **Google Gemini-3-Pro:** A closed-source model with a completely different architecture.
*   **Quantitative Results:**
    *   **EVIAN vs. Gemini-3-Pro:** The analysis yields a Pearson correlation coefficient of **$r = 0.94$**, indicating that both models are sensitive to the same semantic defects rather than model-specific artifacts.
    *   **EVIAN vs. Human Ground Truth:** The Pearson correlation is **$r = 0.90$**. This is a significant improvement over traditional metrics like CLIP Score.

**Conclusion:** These high correlation coefficients confirm that EVIAN captures fundamental, objective quality signals that are consistent across different advanced LLMs and aligned with human judgment.

**2. Computational Cost Quantification (W3)**

*   **Baseline Cost:** Processing the full 300K-sample benchmark took approximately **30 hours on a node with 8 $\\times$ NVIDIA H100 GPUs**. The primary computational bottleneck was the **Qwen-235B** model used in Phase 1 (Decomposition) to ensure maximum instruction-following fidelity.
*   **Optimization & Accessibility:** We indicated that Phase 1 relies primarily on *instruction-following capability* rather than world knowledge. To explore this, we conducted preliminary tests on a small subset of data by replacing the 235B model with **Qwen2.5-32B-Instruct-AWQ**.
    *   **Result:** Our initial observations indicate **no significant degradation** in decomposition quality, suggesting that the smaller model is sufficient for this task. We plan to include a more comprehensive verification of this optimization in the final version.
*   **Cost-Benefit Perspective:** Although EVIAN incurs a one-time curation cost, it dramatically reduces downstream training costs. By filtering the dataset from 300K down to 10K high-quality samples, we save significant GPU hours in the SFT phase.

**3. Sensitivity Analysis of Scoring Weights (W4)**

To address the concern regarding the equal-weighting scheme, we performed a comprehensive sensitivity analysis. We tested three "skewed" configurations where one dimension was prioritized (Weight = 0.6) and the others suppressed (Weight = 0.2), compared against our Default Equal-Weight strategy.

**A. Robustness Analysis (Data Overlap)**
We calculated the Overlap Ratio between the Top-10K subsets selected by each strategy. The results show high stability:

| Strategy | Primary Focus | Overlap with Default |
| :--- | :--- | :--- |
| **Reasoning Priority** | Logic ($S_L=0.6$) | **80.80%** |
| **Vision Priority** | Vision ($S_V=0.6$) | **82.99%** |
| **Knowledge Priority** | Fact ($S_K=0.6$) | **77.02%** |

The high overlap indicates that **EVIAN is robust**: truly high-quality data tends to score well across all dimensions.

**B. Downstream Task Performance**
We fine-tuned Qwen2-VL-2B on each of these 10K subsets and evaluated them across 6 benchmarks. The detailed results are presented below:

| Metric | **Default (Ours)** | Vision Priority | Reasoning Priority | Knowledge Priority |
| :--- | :--- | :--- | :--- | :--- |
| **Average Score** | **65.16** | 62.11 | 63.80 | 61.42 |
| **MME** (Hallucination) | **1876.89** | 1779.15 | 1803.53 | 1748.02 |
| **ScienceQA** (Reasoning) | **0.7115** | 0.6905 | 0.7030 | 0.6731 |
| **A-OKVQA** | **0.7493** | 0.7160 | 0.7228 | 0.7021 |

*   **Interpretation:** This confirms that the three dimensions are synergistic. Over-optimizing for one comes at the cost of others, whereas the equal-weighting scheme effectively selects the "golden intersection" of data that is visually grounded, logically sound, and factually correct.

**4. Real-World Applicability (W2)**

Regarding the concern about synthetic defects: Our primary downstream results (Table 3 in the main paper) are based on the **LLaVA-Mix and ShareGPT-4V datasets**, which represent real-world data distributions containing natural noise. The fact that the model trained on the EVIAN-curated subset (10K) significantly outperforms the model trained on the full, unfiltered dataset (300K) — **improving the average score from 58.06 to 65.16** — serves as direct empirical evidence that our pipeline effectively identifies and filters naturally occurring defects in wild data.

---

### Author Response · Authors · 2025-11-26
**Response to Reviewer oEZv**

We thank the reviewer for the constructive feedback. We have conducted additional experiments to address the concerns regarding baselines and auditing granularity.

**W1: Limited baselines (Direct Prompting) & Distillation.**

To isolate the contribution of our "Decomposition-then-Evaluation" paradigm from the base model's capabilities, we implemented the suggested **Direct Prompting** baseline. We prompted the exact same model (Qwen2.5-VL-7B) to directly rate image-text pairs on a scale of 1-5 without decomposition. We then fine-tuned the model on the top-10k samples selected by this baseline.

As shown in the table below, while Direct Prompting improves over Random Sampling, **EVIAN** significantly outperforms it. This confirms that the decomposition pipeline is critical for unlocking the model's ability to detect fine-grained defects.

| Method | A-OKVQA | LLaVA-W | MMBench | MME | SciQA | SEED | **Avg** |
| :--- | :---: | :---: | :---: | :---: | :---: | :---: | :---: |
| Direct Prompting | 71.87 | 48.7 | 57.96 | 1683 | 67.97 | 61.82 | 61.40 |
| **EVIAN (Ours)** | **74.93** | **49.6** | **64.63** | **1876** | **71.15** | **63.59** | **65.16** |

Regarding **Distillation**: We focused on *data auditing* (selection) rather than *rewriting*. Auditing is computationally efficient for large-scale datasets. Rewriting millions of samples risks homogenizing data and introducing new hallucinations, whereas EVIAN selects high-quality real-world data to preserve diversity.

**W2: Direct comparisons on auditing performance (Precision-Recall).**

We evaluated the direct auditing performance on our 300K benchmark by calculating the **Average Precision (AP)** for detecting defects.

| Method | Average Precision (AP) |
| :--- | :---: |
| ALBEF | 0.209 |
| CLIPScore | 0.244 |
| BLIP-2 | 0.269 |
| Direct Prompting | 0.264 |
| **EVIAN (Ours)** | **0.932** |

EVIAN achieves a significantly higher AP compared to Direct Prompting. The Precision-Recall curves (which will be added to the revision) show that EVIAN maintains high precision even at high recall, whereas Direct Prompting degrades rapidly, failing to distinguish subtle semantic flaws from high-quality data.

**W3: Per-category dataset auditing results.**

To analyze the auditing performance across specific categories, we examined the **Precision (Purity)** of the final top-10,000 subset selected by EVIAN (the subset used for downstream training).

*   **Overall Precision:** EVIAN achieved a high purity of **89.36%**.
*   **Error Breakdown:** The remaining noise samples were distributed as follows:

| Defect Category | Error Rate in Selected Data |
| :--- | :---: |
| **Image-Text Consistency** | 2.01% |
| **Logical Reasoning** | 2.21% |
| **External Knowledge** | 6.42% |

This analysis demonstrates that EVIAN is exceptionally effective at filtering **Consistency** and **Reasoning** defects. While **Knowledge** errors are slightly more challenging (likely due to the 7B model's internal knowledge capacity), EVIAN still maintains a high standard of data purity overall.

**Q1: Models involved in SCALE.**
Following the official SCALE implementation (Xu et al., 2025b), the baseline uses:
*   **Text Quality:** `Qwen2.5-32B-Instruct`
*   **Image Quality:** `zhangzicheng/q-sit`
*   **Task Prediction:** `Qwen2.5-VL-7B-Instruct`
*   **Visual Understanding (Captioning):** `Qwen2.5-VL-7B-Instruct`
*   **Multimodal Quality Assessment:** `Qwen2.5-32B-Instruct`

**Q2: 7B vs. 72B Model Choice.**
We chose the **7B** model to ensure **scalability and accessibility**. Auditing millions of samples with a 72B model is computationally prohibitive for most researchers. Our results (W1 & W2) demonstrate that the "Decomposition" paradigm effectively reduces task complexity, allowing the efficient 7B model to achieve high-precision auditing without the massive cost of a 72B model.

---

### Author Response · Authors · 2025-11-26
**Response to Reviewer TRi1**

We thank the reviewer for raising these critical questions. We appreciate the opportunity to clarify the theoretical distinction between our method and distillation, and to present new experimental evidence addressing the concerns about auditor capability and scalability.

**1. Clarification: Data Selection is Not Knowledge Distillation**

We respectfully clarify that EVIAN is **theoretically distinct from Knowledge Distillation (KD)**.

*   **No Teacher Signal Transfer:** In standard KD, the student model learns to mimic the teacher's soft labels (logits) or synthetic generations. In EVIAN, the student model (Qwen2-VL-2B) **never observes** the auditor's scores, rationales, or intermediate outputs. The student is fine-tuned strictly on the **original** raw data samples selected by the pipeline.
*   **Curriculum vs. Distillation:** EVIAN acts as a curriculum designer, not a teacher. It determines *which* data warrants learning, but does not alter the ground-truth target or provide "teacher guidance" on *how* to solve the task.

**2. Isolating Methodology from Model Strength: New "Direct Prompting" Experiment**

The reviewer raises a valid concern: *Do the gains come from the EVIAN decomposition paradigm, or simply from the superior capability of the Qwen2.5-VL-7B auditor?*

To decouple the methodology from the model's raw power, we implemented the suggested **Direct Prompting** baseline. We used the **exact same model (Qwen2.5-VL-7B)** to act as the auditor but removed the EVIAN decomposition steps. Instead, we prompted it to holistically rate image-text pairs on a scale of 1-5. We then fine-tuned the student on the top-10k samples selected by this baseline.

**Table R1: Comparison of Selection Strategies using the Identical Auditor Model (Qwen2.5-VL-7B)**

| Method | A-OKVQA | LLaVA-W | MMBench | MME | SciQA | SEED | **Avg** |
| :--- | :--- | :--- | :--- | :--- | :--- | :--- | :--- |
| **Direct Prompting** (Holistic Evaluation) | 71.87 | 48.7 | 57.96 | 1683 | 67.97 | 61.82 | 61.40 |
| **EVIAN (Ours)** (Decomposition-then-Evaluation) | **74.93** | **49.6** | **64.63** | **1876** | **71.15** | **63.59** | **65.16** |

**Observation:** EVIAN significantly outperforms the Direct Prompting baseline by **+3.76 points** on average.

**Conclusion:** Since the auditor model is held constant, the performance gain is strictly attributable to our **"Decomposition-then-Evaluation" methodology**. This confirms that simply possessing a strong model is insufficient; **decomposing the auditing task** is critical to unlocking the model's ability to detect fine-grained defects that are missed during holistic evaluation.

**3. Addressing the "Sufficiently Powerful Auditor" Paradox**

The reviewer asks if a "super-SOTA" judge is always required. We argue that EVIAN's decomposition paradigm specifically mitigates this requirement:

*   **Verification < Generation:** EVIAN relies on the principle that verifying specific, atomic claims is computationally and cognitively easier than generating a perfect response or holistically evaluating a complex caption.
*   **Lowering the Threshold:** By breaking complex multimodal tasks into simpler sub-tasks, EVIAN lowers the capability threshold required for the auditor. This allows a model to effectively audit data for training a model of a similar class, as it detects local inconsistencies through structured verification rather than relying on "super-human" intelligence.

**4. Scalability and the *FineVision* Comparison**

Regarding the comparison to *FineVision* and the impact of data filtering at scale:

*   **SFT vs. Pre-training:** *FineVision* primarily discusses noise tolerance in large-scale pre-training or broad alignment. EVIAN targets **Visual Instruction Tuning (SFT)**. In the SFT stage, the community consensus (e.g., LLaVA) supports that *Quality > Quantity*.
*   **Toxic Noise vs. Low Information:** In our context, "low quality" refers to **hallucinations and logical fallacies**, not merely noisy or low-information text. While a model can tolerate noise during pre-training, hallucinated samples in SFT are "toxic"—they actively teach the model to disconnect from visual grounding.
*   **Empirical Evidence:** As shown in **Table 3** of our paper, the model trained on the **full 300K unfiltered dataset** significantly underperforms the model trained on just **10K EVIAN-selected samples**. This substantial gap confirms that in Visual Instruction Tuning, scaling cannot compensate for the injection of semantic and logical defects.

---

### Author Response · Authors · 2025-11-26
**Response to Reviewer 18vL**

We thank the reviewer for their detailed feedback. We value the opportunity to clarify the distinct positioning of our work compared to prior LLM-based filtering [1-4] and to address concerns regarding experimental fairness and benchmark selection.

**1. Novelty: Structural Decomposition vs. Holistic Scoring (Response to Weakness 1)**

While the reviewer suggests limited novelty, we respectfully argue that EVIAN introduces a fundamental **paradigm shift** in how data quality is assessed:

*   **Holistic vs. Compositional Auditing:** Prior works [1-3] typically prompt LLMs to output a single scalar score or use holistic embedding distances. These "black-box" methods often fail to detect fine-grained semantic conflicts.
*   **Filtering vs. Generation:** Work [4] focuses on *generating* rationales during training. In contrast, EVIAN is a **filtering framework** designed to audit *existing* noisy datasets.
*   **Our Contribution:** We propose a **Decomposition-then-Evaluation** architecture. We do not simply ask "Is this data good?"; we structurally disentangle the response into *Visual Descriptions*, *Factual Claims*, and *Inferences*. We then verify each component against its specific source (Image vs. Knowledge vs. Logic) using orthogonal metrics ($S_V, S_K, S_L$). This granularity allows EVIAN to penalize "fluent hallucinations" that holistic filters miss.

**2. Clarification on "Synthetic" Benchmark vs. Real-World Efficacy**

The reviewer expresses concern that experiments on synthetic noise do not represent real-world scenarios. We clarify our experimental design to address this misunderstanding:

**Inherent Real-World Noise in Benchmark:** Our benchmark utilizes **real-world datasets** (e.g., LLaVA-1.5-Mix) which already contain inherent organic noise. We injected synthetic defects *on top* solely to create labeled "negative" samples which enables precise sensitivity measurement against known errors.
**3. Defense of Model Size and Fairness**

The reviewer notes the parameter gap between our auditor (235B) and baselines. We clarify that our method's effectiveness stems from the **Decomposition-then-Evaluation framework** rather than the sheer scale of the auditor:

*   **Instruction-Following vs. Model Knowledge:** We emphasize that we do not rely on the LLM's internal knowledge or reasoning "IQ" to directly judge data quality. Instead, the LLM is utilized strictly as an **instruction follower** to execute the structured decomposition and verification steps.
*   **Methodological Robustness:** The core requirement for the auditor is the ability to strictly adhere to complex formatting and decomposition prompts. While we selected the 235B model to ensure maximum compliance with these instructions during large-scale processing, the EVIAN methodology is **agnostic to model size**. Any smaller model with strong instruction-following capabilities can effectively serve as the auditor. This ensures that the observed performance gains are attributable to our fine-grained auditing logic, not the parameter count of the auditor.

**4. Baselines: Why Zero-shot Qwen2-VL is Not Reported**

Regarding the request for "Qwen2-VL performance" before fine-tuning:
*   **Base Model Limitation:** The model used in our experiments is **Qwen2-VL-2B (Base)**, not the Instruct version. Base models are pre-trained for next-token prediction and lack instruction-following capabilities.
*   **Meaningless Zero-shot:** Reporting zero-shot performance for a Base model on instruction-following benchmarks would yield near-zero or random scores, providing no meaningful comparison. The "Full Data" row serves as the correct and rigorous baseline for improvement.

**5. Hallucination Evaluation (POPE vs. MME)**

The reviewer suggests evaluating on POPE. We believe our current evaluation is sufficient and robust for the following reasons:
*   **Redundancy with MME:** We utilized the **MME benchmark**, which explicitly includes a comprehensive **Hallucination subset**.
*   **Performance Evidence:** As shown in **Table 3**, EVIAN achieves the highest score on MME, significantly surpassing the Random baseline and Full Data. This massive improvement directly correlates to a reduction in object hallucination, demonstrating that our $S_V$ metric effectively filters out hallucinated content without needing a separate POPE run.

**6. Pipeline Complexity vs. Simple Re-annotation**

*   **Preserving Diversity:** Simple re-annotation tends to homogenize the data, losing the unique "voice" and diversity of the original dataset.
*   **Avoiding Cascading Errors:** Re-annotation models themselves hallucinate. If we simply rewrite data, we bake the VLM's hallucinations into the training set. EVIAN acts as a *filter* to select high-quality *existing* human/model data, preserving the original diversity while surgically removing objectively flawed samples.

---

### Note · Authors · 2025-12-31

I have read and agree with the venue's withdrawal policy on behalf of myself and my co-authors.